# Mordenite-Supported Ag^+^-Cu^2+^-Zn^2+^ Trimetallic System: A Variety of Nanospecies Obtained via Thermal Reduction in Hydrogen Followed by Cooling in an Air or Hydrogen Atmosphere

**DOI:** 10.3390/ma16010221

**Published:** 2022-12-26

**Authors:** Inocente Rodríguez-Iznaga, Vitalii Petranovskii, Felipe F. Castillón-Barraza, Sergio Fuentes-Moyado, Fernando Chávez-Rivas, Alexey Pestryakov

**Affiliations:** 1Instituto de Ciencia y Tecnología de Materiales (IMRE), Universidad de La Habana, Zapata y G, La Habana 10400, Cuba; 2Centro de Nanociencias y Nanotecnología, Universidad Nacional Autónoma de México, Ensenada 22800, Mexico; 3Departamento de Física, ESFM, Instituto Politécnico Nacional, UPALM, Zacatenco, Ciudad de México 07738, Mexico; 4Research School of Chemistry and Applied Biomedical Sciences, Tomsk Polytechnic University, Tomsk 634050, Russia; 5Laboratory of Catalytic and Biomedical Technologies, Sevastopol State University, Sevastopol 299053, Russia

**Keywords:** mordenite zeolite, trimetallic system, copper, silver, zinc, thermal redox, nanospecies

## Abstract

Multimetallic systems, instead of monometallic systems, have been used to develop materials with diverse supported species to improve their catalytic, antimicrobial activity, etc., properties. The changes in the types of nanospecies obtained through the thermal reduction of ternary Ag^+^-Cu^2+^-Zn^2+^/mordenite systems in hydrogen, followed by their cooling in an air or hydrogen atmosphere, were studied. Such combinations of trimetallic systems with different metal content, variable ratios (between them), and alternating atmosphere types (during the cooling after reducing the samples in hydrogen at 350 °C) lead to diversity in the obtained copper and silver nanospecies. No reduction of Zn^2+^ was evidenced. A low silver content leads to the formation of reduced silver clusters, while the formation of nanoparticles of a bigger size takes place in the trimetallic samples with high silver content. The cooling of the reduced trimetallic samples in the air causes the oxidation of the obtained metallic clusters and silver and copper nanoparticles. In the case of copper, such conditions lead to the formation of mainly copper (II) oxide, while the silver nanospecies are converted mainly into clusters and nanoparticles. The zinc cations provoked changes in the mordenite matrix, which was associated with the formation of point oxygen defects in the mordenite structure and the formation of surface zinc oxide sub-nanoparticles in the samples cooled in the air.

## 1. Introduction

The modification of zeolites with metallic nanospecies represents an important approach to developing materials for various purposes. The crystalline structure of the zeolites has channels and cages that are of nanometer or sub-nanometer in size, which are able to confine ions, clusters, and nanoparticles. This is crucial for the development of metal/zeolite materials with important properties, such as catalytic and antimicrobial activity. Metals, such as copper, zinc, and silver, as well as their compounds, are of great interest due to both their catalytic properties [1,2,3,4,5,6] and oligodynamic activity [7,8,9,10,11].

Depending on the initial composition and conditions of the preparation process, a huge variety of nanospecies can be formed on zeolite supports; cations, charged and/or neutral small clusters, subcolloidal and colloidal species, nanoparticles, and large particles, as well as a mixture of these species, which are widely reported as substances supported on zeolite matrices.

The useful properties of complex materials are determined by the presence of certain compounds within them. The properties of metal/zeolite materials depend on, among other aspects, the type of the implanted species, their size, and their oxidation state.

The processes of preparing the zeolite supports that are modified with metallic nanospecies, as a rule, begins with ion-exchange. The prepared zeolites are then subjected to subsequent treatment, where reduction is often used to produce metallic nanoparticles. Reports are mainly known for the monometallic systems obtained by the conventional ion-exchange procedure. Such kinds of studies are generally performed using a thermal reduction in a flow of hydrogen, followed by cooling in the same reducing agent. However, an important aspect is the type of atmosphere that was used during the initial heating, reduction, and subsequent cooling of the samples. Varying these conditions can lead to the formation of nanospecies that differ significantly in their structure and properties [12,13,14,15,16].

Important improvements in the properties of nanospecies have been found in multimetallic zeolite-supported materials [17,18,19,20,21]. In polymetallic systems, when compared to monometallic ones, there are considerable differences in the kinds of metal nanospecies obtained by thermal reduction, which, of course, drastically alters the properties of the resulting materials. Thermal reduction in a hydrogen flow for the bimetallic Cu^2+^-Ag^+^ and Cu^2+^-Zn^2+^ systems supported on both natural clinoptilolite [18,19,21] and mordenite [20] showed the mutual influence of the Cu^2+^ and Ag^+^ cations on the reduction process, and the formation of various nanospecies for both elements. The reduction of Cu^2+^ is facilitated by the presence of Ag^+^ and Zn^2+^, mainly at low temperatures. The aggregation of reduced silver and copper in these bimetallic systems is limited compared to monometallic systems (Cu-only or Ag-only zeolites). Studies on the long-term changes of reduced copper-silver bimetallic systems show that silver affects the decomposition/oxidation mechanism and increases the stability of the reduced copper species [21].

The role of co-cations in multimetallic materials can be different. It was shown that the addition of silver as a co-cation in the Cu-ZSM-5 catalyst used in the NO SCR process increased resistance to dealumination, which was induced by acids formed as a result of reactions between NO, O_2_, and H_2_O, and simultaneously decreased the reversible inhibition by water, which increased the hydrothermal stability under an air–H_2_O mixture [22]. Ramírez et al. [6] also reported higher hydrothermal stability and catalytic activity in NO reduction using bimetallic Cu-Ag/Mordenite catalysts compared to monometallic Cu/Mordenite versions. This was associated with the fact that silver facilitates the Cu^+^/Cu^2+^ redox cycle during NO reduction, and it acts as a shield to protect the copper species by coordinating with the water molecules under hydrothermal conditions. Sánchez-López et al. reported that catalytic activity in NO reduction was higher for bimetallic FeAg/mordenite catalysts than for Fe/Mordenite versions [23].

Song et al. [24] studied the deep desulfurization of gasoline over bimetallic Cu–Ce ion-exchanged Y zeolite. They reported that Cu^I^Ce^IV^Y not only has a high sulfur adsorption capacity similar to that of Cu^I^Y but also has high selectivity for sulfur compounds similar to Ce^IV^Y. The high selectivity of Cu^I^Ce^IV^Y for sulfur removal was attributed to a synergistic interaction between Cu^+^ and Ce^4+^. Khatamian et al. [25] reported that the Fe-TiO_2_/ZSM-5 composite is a more efficient catalyst for the degradation of dye molecules when compared with TiO_2_/ZSM-5. This effect is attributed to the effective separation of the charge carriers by the Fe^3+^ ions acting as traps for both electrons and holes. Kim et al. [26] reported the selective catalytic reduction of NOx over catalysts prepared on γ-Al_2_O_3_ monoliths and NaY. Ag, Cu, Pt, and Co modifiers were employed in a double-layer or dual-bed configuration or a combination thereof. Among the catalysts tested, a trimetallic dual-bed monolith catalyst was identified to be the best combination for maximum NOx-to-N_2_ conversion, with the minimum formation of harmful byproducts; this was explained through multifunctional cooperative catalytic processes with Ag/Al_2_O_3_, CuCoY, and Pd/Al_2_O_3_. Catalysts prepared from natural mordenite modified with Ni, Ni-Mo, Co, and Co-Mo species were used for fuel production from LDPE plastic waste [27]. The result of the hydrocracking experiment shows that the conversion of Co-Mo/mordenite at 350 °C demonstrates higher selectivity to gasoline. The authors proposed that the improvement of catalytic effects on this bimetallic catalyst was due to the acidic properties that have been explained by a carbonium ion mechanism.

Studies of linde type A zeolite modified with Cu^2+^, Ag^+^, and Zn^2+^ have shown that bimetallic AgCu-linde and AgZn-linde systems have higher antimicrobial activity (against *E. coli*) compared to monometallic Ag-linde systems, which may lead to the development of composite zeolite containing antimicrobial materials of a new generation for different biomedical and pharmaceutical uses [11].

The aforementioned studies on zeolites modified with multimetallic systems, mostly bimetallic, have shown that the second co-cation has a significant influence. In a number of cases, improvements to the properties of the resulting materials were obtained. The use of more complex systems could then increase the synergistic interaction, which can be important for the development of new materials. Thus, trimetallic copper, zinc, and silver systems are of interest for improving catalytic and biocide activity when compared to monometallic versions, which are supported on zeolites in the form of mixed cationic systems and are then subjected to reduction processes, followed by different cooling conditions (starting with a temperature higher than room temperature) that can lead to the obtainment of a wide variety of nanospecies and materials with a broad spectrum of properties. In addition, the preparation of these materials commonly begins with ion exchange processes under hydrothermal conditions, applied over hours and days, i.e., in the traditional way. However, there are other methods that allow the ionic exchange time to be reduced, such as using microwave radiation [28].

In this work, changes in the metallic type of the nanospecies obtained by thermal reduction in hydrogen, followed by cooling in an air or hydrogen atmosphere for the trimetallic Ag^+^-Cu^2+^-Zn^2+^ system supported on mordenite through microwave-radiation assisted ion-exchange processes were studied.

## 2. Materials and Methods

Synthetic Na-mordenite (NaMor) with a SiO_2_/Al_2_O_3_ molar ratio equal to 13 was supplied by Zeolyst International. Starting material was exchanged under microwave-radiation action with 0.5 N solutions of AgNO_3_, Cu(NO_3_)_2_, Zn(NO_3_)_2_, and their mixtures, using a Synthos 300 Anton Paar oven (Media, PA, USA)equipped with a microwave reaction system. All used salts (AgNO_3_, Cu(NO_3_)_2_, and Zn(NO_3_)_2_) were reactive grade; they were supplied by Sigma-Aldrich, St. Louis, MO, USA. The mixed solutions used to obtain the ternary samples were prepared by mixing of AgNO_3_, Cu(NO_3_)_2_, and Zn(NO_3_)_2_ solutions in 1:1:1, 1:1:2, 1:2:1, 1:2:4, 1:2:6, and 1:2:8 ratios, respectively. The different ratios were established to obtain samples with diverse Cu, Zn, and Ag contents; it was made taking into account the ion exchange capacity of this mordenite and the knowledge that most of zeolites have higher selectivity by Ag^+^ cations [29]. These relationships were used for the further labeling of trimetallic samples as the numbers after symbol of each metal. Solid/solution ratio was maintained, equal to 1 g/10 mL. The exchange was performed at a 100 °C constant temperature during 120 min with a ramp 15 °C/min.

The resulting monometallic (AgMor, CuMorand ZnMor) and trimetallic (AgCuZn/Mor) samples were heated in H_2_ flow at 350 °C over 4 h. After the treatment, the reduced samples were cooled to room temperature, applying two different atmospheres: the same H_2_ flow, or air. The reduced and cooled in H_2_ samples were labeled by “hc” (“hydrogen-cooled”, for example CuMor-hc), while the reduced and cooled in air were labeled by “ac” (“air-cooled”, for example CuMor-ac).

The two sets of prepared samples were studied by UV-Vis diffuse reflectance (UV-Vis) spectroscopy and X-ray diffraction (XRD). UV-Vis spectra were obtained with a Perkin Elmer 330 spectrometer (Richmond, CA, USA) equipped with a standard diffuse reflectance unit using barium sulfate as a reference. XRD patterns were recorded in a Philips X’Pert diffractometer (Almelo, The Netherlands) equipped with a curved graphite monochromator, using CuK_α_ (λ = 0.154 nm) radiation. Composition of the samples was determined by the EDS (energy dispersive spectroscopy) method using a JEOL 5200 scanning electron microscope (Tokyo, Japan) equipped with a Kevex Super Dry EDS apparatus. Quantification was carried out using the standard Magic 5 software.

## 3. Results and Discussions

### 3.1. Composition and Characterization of the Exchanged Samples

The composition of the exchanged samples is presented in Table 1. In order to represent the compositions of our ternary mixture, we used a triangular diagram (Figure 1), each vertex of which corresponds to one of the metallic components. The concentrations (or contributions) of these components are plotted on the sides of the triangle. In Figure 1, the circles show the percentages of the components in the composition of the initial ternary solutions (e.g., circle 2: 25% Ag, 25% Cu, and 50% Zn in the initial mixed solution) used to prepare the trimetallic samples, while the squares show the percentages of the metals supported on the trimetallic samples (e.g., square 2: 72.73% Ag, 15.36% Cu, and 11.90% Zn) prepared from these solutions as a result of the experiment. When analyzing this diagram, it can be concluded that the selectivity of mordenite for Zn^2+^ is lower than that for the Cu^2+^ and Ag^+^ cations. Although the amount of zinc (%) in the exchanged trimetallic samples increases with increasing zinc solution content in the mixture, these amounts are disproportionately smaller when compared to the other two cations. Note that the values of the ratio of the volumes of the starting ternary solution (see circles in Figure 1) are closer to the maximum of the zinc content than for the other metals (copper and silver), while the ratio of the amounts applied to mordenite (see squares in Figure 1) is further away from the maximum zinc exchangeable. In general, the Cu^2+^ content did not differ significantly between the exchanged trimetallic samples (see Table 1). Besides this, mordenite showed a higher percentage of metal uptake for silver; it can be noted that the ratio of the quantities (see squares in Figure 1) is closer to the maximum silver content. This observation correlates with the known results, according to which most of the zeolites exhibit the highest affinity for the Ag^+^ cations in the ion exchange processes [29].

Heavy metal cations appear in the zeolite matrices as a result of ion exchange between the Na^+^ cations from NaMor and the Ag^+^, Cu^2+^, and Zn^2+^ cations from the used exchange solutions. In this study, a method of ion exchange under microwave radiation was applied to increase the metal content in the samples, as well as to decrease the ion-exchange time compared to the conventional exchange method. Indeed, the amounts of Cu^2+^ and Zn^2+^ cations in the samples prepared via the exchange treatment assisted with microwave radiation are superior to those achieved with the traditional exchange, particularly for zinc [17]. This allows us to conclude that the accessibility of the cationic sites for the Cu^2+^ and Zn^2+^ ions increases in the case of an exchange assisted by microwave radiation. Increasing the amount of deposited metal can be important in the case of the intended practical applications of these materials as, for example, catalysts, fungicides, or microbicides. This represents an important method for producing materials with a wide range of properties based on a synergy of the qualities of all three different metals (silver, copper, and zinc) that are present simultaneously.

The UV-Vis spectrum (Figure 2) of the exchanged CuMor sample shows a characteristic band of the Cu^2+^ cation, centered at 800–850 nm, and a charge-transfer band at ~210 nm due to the interaction of the Cu^2+^ cations with the oxygen of the zeolite framework. The Ag^+^ and Zn^2+^ cations themselves do not have characteristic absorbance in the UV-Vis range. However, similarly to Cu^2+^, these last two cations interact with the zeolitic framework, and then both experience a charge-transfer complex, the strength of which, as should be expected, is reflected in the absorbance intensity in the spectra. Note that, in the absorption spectrum of the AgMor sample prepared by microwave-assisted exchange, an intense band in the region of 200–210 nm is observed (Figure 2). With a high degree of probability, this band is the tail of the 4d^10^–4d^9^5s^1^ transition band of the Ag^+^ ion, centered in the vacuum ultraviolet [30,31,32], which is outside the range of the spectral sensitivity of the spectrometer used in this work. This assumption is confirmed by the data reported in reference [33] about the UV-Vis spectra of AgZSM-5 samples with variable Ag content. It was found that the absorption of light at a wavelength of 200 nm considerably increases, while the amount of silver in the sample changed from 0.5 to 3.9 wt.%. Nevertheless, this issue goes beyond the scope of the proposed study and will be reviewed and published elsewhere. Since this band in the trimetallic sample overlaps with the band observed in the CuMor sample, it is impossible to unambiguously attribute the observed absorbance in this range, and its resulting intensity should be linked to the contribution of all these cations.

The XRD pattern of the NaMor sample, as well as the patterns of the ion-exchanged monometallic and trimetallic samples, are shown in Figure 3. The data show that there are no significant changes in the crystal structure of mordenite during the ion exchange treatment, indicating that the use of microwave radiation does not affect the structure of this zeolite. The only observable effect is a variation in the relative intensities of individual peaks for the exchanged samples. These changes are common and consistent with the known observations of ion-exchanged zeolites. They are fundamentally associated with differences in the nature, quantity, and position of extra-framework ions in the mordenite channels [18,34]. The incorporation of exchangeable cations in the channels of mordenite affects the diffraction peaks in accordance with the extra-framework species positioned in a certain mirror plane. So, the variation in the intensities of the *(200)* and *(110)* peaks are strongly related to the extra-framework species positioned in the mirror plane perpendicular to the *a*-axis and the mirror plane formed by *a* and *b* axis, respectively. It is important to note that the intensity of the *(200)* peak decreases with increasing silver content in the samples. This permits us to suggest that the Ag^+^ ions occupy a different cationic position with respect to the starting Na^+^ cations in mordenite.

### 3.2. Thermal Reduction of the Exchanged Samples Followed by Cooling in either Air or Hydrogen

The X-ray diffraction patterns of the samples reduced and cooled in hydrogen (labeled as -hc) are shown in Figure 4. In the patterns of all the reduced trimetallic samples, new intense peaks at 38.11° and 44.29°, which correspond to metallic silver, are visible, which is in line with peaks at 43.2° and 50.3°, corresponding to metallic copper. The monometallic CuMor-hc and AgMor-hc samples also show these peaks for copper and silver, respectively. The results indicate that the reduction of the Cu^2+^ and Ag^+^ cations by H_2_ under the applied conditions and the subsequent agglomeration of the neutral atoms results in copper and silver metal particle formation. Note that the intensity of the *(200)* peak increases in the AgMor-hc and trimetallic samples (reduced and cooled in hydrogen), when compared to the same unreduced exchanged samples (Figure 3). This was associated with the absence of the Ag^+^ ions in the reduced samples that contain silver (Figure 4). It is opportune to remember that the *(200)* peak intensity in these exchanged samples (Figure 3) was low, and such intensity decreases with increasing silver content.

XRD patterns of the samples reduced in hydrogen and then cooled in air (labeled as -ac) are shown in Figure 5. The patterns of the monometallic AgMor-ac and trimetallic ac samples show peaks at 38.1° and 44.2° corresponding to metallic silver, but the intensity of both the peaks is lower with respect to the AgMor-hc and trimetallic hc samples (reduced and cooled in hydrogen). This demonstrates that the amount of silver nanoparticles decreases with respect to the samples reduced and cooled in hydrogen. In the case of CuMor-ac, the peaks associated with metallic copper are not visible on its XRD pattern. However, a new weak peak emerges at 35.6°, and a broad diffraction line is observed centered at 38.7°; these changes in a more detailed image are visualized in Figure 6. Both diffractions were attributed to copper (II) oxide, according to [1,12,35]. It is valid to emphasize that the diffraction lines are significantly broader, which could be due to the very small size of obtained CuO crystallites. The diffraction patterns of the trimetallic ac samples also show the 35.6° peak associated with CuO (Figure 5 and Figure 6); however, the 38.7° broad diffraction is not observed in these samples, which might be due to the low copper content with respect to CuMor-ac and the overlap of the silver peak at 38.11°. The formation of CuO was linked to the reaction between the reduced copper nanospecies and the oxygen from the airflow during the cooling process of the samples.

The UV-Vis spectra of the hc and ac samples are shown in Figure 7 and Figure 8, respectively. The UV-Vis spectra of all the trimetallic hc samples show a broad optical absorption with asymmetric peaks at 280–323 nm and ~400 nm, which is in line with a band at 550 nm. This last band is due to the plasma resonance of nanometer-sized colloidal copper particles, which is also present in the CuMor-hc monometallic sample. The absorption at 280–325 nm and ~400 nm is related to silver; the latter peak refers to the surface plasmon resonance of silver nanoparticles. These results are consistent with those obtained by XRD for the formation of copper and silver metals by hydrogen reduction and are consistent with previous studies [18,36,37,38,39]. The absorption peaks in the 280–325 nm range are associated with silver clusters, which agrees with the data reported in [18,21,39,40,41,42]. In the case of AgMor-hc, peaks centered at 290 nm and 320 nm are observed. The latter peak is assigned to a neutral Ag_8_ cluster [18,21,39,40,42], while the peak at 290 nm has been linked to positively charged clusters of different nuclearities, Ag_n_^δ+^ (n < 10) [41,43]. For the Ag_1_Cu_2_Zn_6_Mor-hc and Ag_1_Cu_2_Zn_8_Mor-hc samples, which have a low silver content (see Table 1), the absorption peaks of the silver species have a low intensity; however, in their spectra, there is a band at 320 nm associated with neutral Ag_8_ clusters [13,37,38,39], which was not observed in other trimetallic samples. The results indicate a reduction of the Cu^2+^ and Ag^+^ cations under the applied conditions. In addition, this observation shows that, in the trimetallic samples, the formation of the Ag_8_ clusters correlates with a lower silver content (see Table 1), while the formation of the reduced silver species of larger sizes takes place in the trimetallic samples with higher silver content.

In the case of all the ac samples, the UV-Vis spectra (Figure 8) show that the absorption at 280–325 nm and ~400 nm has low intensity. This indicates a decrease in the number of silver nanoparticles compared to the hc samples. In addition, there are no data on the presence of reduced copper nanoparticles, which were observed in the spectra of the hc samples. In the spectra of the ac samples, instead of the 550 nm band that is characteristic of the plasma resonance of nanometer-size colloidal copper particles, a broad optical absorption that extends to the visible range is observed. This was associated with the oxidation of the reduced nanospecies due to their reaction with the oxygen from the airflow during the cooling process. Additionally, there are bands at ~210 nm, which were not observed in the spectra of the hc samples. These bands at 210 nm are due to the charge-transfer complex caused by the interaction between the cationic species (Cu^2+^, Ag^+^, and Zn^2+^) and the oxygen of the zeolite framework. These findings indicate that cooling in the air causes the oxidation of the reduced clusters and nanoparticles, which is in line with the results obtained by XRD. This oxidation process is kinetically favored due to the applied temperature (350 °C), resulting in the formation of mainly copper (II) oxide, which should be accompanied by the formation of small amounts of ionic species, such as Cu^2+^ and Cu^+^.

In the case of silver, the resulting species are clusters, nanoparticles, and Ag^+^ cations. However, the oxidation of Ag^0^ to Ag^+^ should be limited. Note that the intensity of the (*200*) peak of mordenite is higher in the AgMor-ac and trimetallic ac samples than in the unreduced exchanged samples. Signs of the formation of silver oxide were not found. Nevertheless, in the catalysts based on different amounts of silver deposited on mordenite (5, 10, and 15 wt.% Ag, Ag_x_MOR), an EXAFS analysis of the samples treated in oxidizing (O_2_) or reducing (H_2_/Ar) atmospheres showed interesting features [43]. In the calcined in oxygen samples, the EXAFS profiles revealed two types of Ag-O co-ordination spheres: one, as expected, due to ion exchange Ag^+^ cations due to the interaction with the oxygen of zeolite framework, and the other one is attributed by the authors to a dispersed silver oxide phase. At the same time, the DRS spectra of these samples in the UV-visible region [43] showed the coexistence of isolated silver cationic clusters (Ag_n_^δ+^ (n < 10)) and Ag_8_neutral clusters. These clusters were also observed in our work (absorbance at around 290 and 320 nm, Figure 7 and Figure 8). The chemistry of the silver deposited on the zeolites in the case of the monometallic samples is discussed in more detail in the review [44].

No evidence of the Zn^2+^ reduction for the hc samples was found. However, the intrinsic spectrum of the mordenite matrix changed after this treatment. The spectra of the monometallic zinc-containing samples subjected to various treatments (in the 200–500 nm range) show changes in the intensity and position of the absorbance bands in the UV region (Figure 9). In the case of the monometallic zinc-mordenite, the changes in the optical density of the samples, as compared to the initial mordenite, were greater when the samples were cooled in the air (see Figure 9) than when they are cooled in hydrogen. As a result, the intensity of the absorption bands (see Figure 9) for ZnMor-ac increases with respect to both ZnMor (exchange sample) and ZnMor-hc. It can be observed that the band centered at 225 nm in the exchanged ZnMor sample changes slightly and appears centered at 237 nm in both ZnMor-hc and ZnMor-ac. However, the shape of the absorption peaks for ZnMor-ac differs from that of ZnMor-hc and is similar to that in the zeolites with supported zinc oxide [45].

All of the above leads to the consideration of the possibility of sub-nanodisperse zinc oxide formation in the Zn-containing samples reduced in hydrogen and then cooled in the air, for which the size and/or concentration are insufficient for the appearance of a signal in XRD. Unlike silver and copper, the fact that zinc is not reduced in hydrogen under these conditions also explains the difference in the behavior of the monometallic samples in their contact with reaction temperatures in air/oxygen. When reduced in hydrogen, the metallic cations of silver or copper are replaced by protons, and the general equilibrium between the negative charge of the framework and the pool of cations is maintained. Then, upon contact with oxygen, the reduced metallic varieties are oxidized, and this reaction does not affect the crystalline structure of mordenite. In the case of zinc-mordenite, the remaining unreduced cations cannot form zinc oxide upon contact with oxygen because any compounds that could take on the role of cations that balance the charge of the zeolite framework are absent. Therefore, the only reaction that can theoretically take place is the formation of point oxygen defects at the locations of the double-charged zinc cations in the vicinity of the two Al-containing tetrahedrons of the zeolite structure. Perhaps it is the formation of such defects that is responsible for the observed increase in adsorption in the UV region.

These weak and broad peaks in the UV region, which change during the exchange and heat treatment in hydrogen and become even stronger after the samples make contact at 350 °C with air oxygen, show the processes occurring in the mordenite matrix. In the case of the polymetallic samples, the processes of the silver and copper ions, in contrast to monometallic samples, proceed on a matrix modified by the very presence of the zinc ions. This influence can be reflected both in the formation of the reduced and oxidized species as well as in their further behavior.

Undoubtedly, the preparation of the nano-dispersed particles of semiconductor zinc oxide is of considerable interest for the creation of photocatalysts [46,47] or compounds with oligodynamic properties [48,49]. There is an extensive range of methods for such syntheses; for example, the preprecipitation of zinc hydroxide via treatment in alkaline solutions with a controlled pH, followed by calcination to prepare ZnO [50,51]. Zinc ions can be reduced to a metallic state by reduction treatment at temperatures above 550 °C [52]. However, under the process conditions chosen for this work, such reactions cannot occur. Studies in a wider range of reaction conditions are planned; their results will be published elsewhere.

Several studies are available in the literature on the formation of copper, zinc, and silver oxides in zeolites and other matrices, but for those samples prepared by ion-exchange or impregnation methods followed by thermal treatment in the air [1,2,3,7,35,53,54]. The results on the oxidation of reduced silver and copper nanospecies under elevated temperatures obtained in this work are different in comparison with the reported earlier data [23,25,26] on the decomposition of other samples reduced and cooled in hydrogen observed during storage under ambient conditions. In the latter case, the reduced metallic nanospecies existing in the fresh samples were reoxidized into ionic species (e.g., Cu^2+^ cation); or smaller nanospecies (e.g., cluster, nanoparticles, etc.) and appeared as intermediates before the final, complete oxidation, and no evidence exists on the formation of metal oxides.

Ultimately, both the synthesis and the further behavior of the reduced copper nanospecies are determined by the conditions in which they are located. During the initial reactions (see Equation (1)), the reduced copper atoms agglomerate into both clusters, for which there is enough space in mordenite channels with a diameter of about 0.7 nm, and into larger diameter nanoparticles on the surface of mordenite crystals (Equation (2)).
Cu^2+^MOR + H_2_ → Cu^0^@(H^+^)_2_MOR(1)
nCu^0^ → (Cu^0^)_n_@(H^+^)_2_MOR(2)

The “@” symbol is used to denote species that are related to the zeolite matrix only by their location because they were formed in its voids. In this case, mordenite itself, after cooling to room temperature and rehydration, acquires acidic properties since the role of exchange cations is performed by protons (H^+^ in a single representation hydronium cation).

When cooled in a reducing atmosphere, the nanospecies of metallic copper that are formed on the surface and in the voids of mordenite crystals are surrounded by water adsorbed by zeolite after contact with the atmosphere. As a result, at room temperature, they can interact with these protons, with the usual reverse reaction of (Cu^0^)_n_ and (H^+^)_2_MOR then occurring and the subsequent spreading of the formed copper ions over the cation-exchange positions of mordenite (assisted by water), which is located in the channels of the zeolite. This process is kinetically inhibited under environmental conditions, and its results clearly manifest only during long periods of storage [21].

In the case of the present work, when the reducing atmosphere was replaced by an oxidizing one at the temperature of the reduction process (350 °C) for the formed copper species, both the nanoparticles on the surface of mordenite crystals and the metal clusters in channels interact with atmospheric oxygen and oxidize to copper oxide (Equation (3)).
(Cu^0^)_n_@(H^+^)_2_MOR + n/2O_2_ → (CuO)_n_@(H^+^)_2_MOR(3)

At the same time, they are also located on protonated mordenite, such that, with a very high degree of probability, after long-time storage, these samples will also revert to the ionic copper form similar to the process of the metallic copper particles reported in [21]. However, before this happens, the resulting copper oxide nanoparticles and clusters can be put to good use.

The direct conversion of methane to methanol (DMTM) is an efficient and practical process for improving natural gas efficiency. Studies in recent years have shown that Cu oxo-nanoclusters placed in microporous solids, which have the ability to activate CH_4_ at moderate temperatures, proved to be promising catalysts for DMTM. The activity of such materials in the selective oxidation of methane to methanol is discussed in a review article [55]; it compiles the possible mechanisms for this reaction, taking into account the different structural states of active Cu-oxo centers.

Among the different Cu-zeolites that have been investigated for the DMTM process, Cu-mordenite has been one of the most active [56]. Many different catalysts for achieving high methane-to-methanol conversions have been studied in recent decades, including Cu-based enzymes, Cu-zeolites, Cu-MOF (metal-organic frameworks), and Cu-oxides. A review [57] outlines the findings on the exact state of the Cu-active centers for these various catalysts, which arose from the most recently developed methods and the results of DFT calculations. The relationship between structure and its characterization in terms of the properties of these materials and their catalytic functions are the main problems that remain.

Varieties of di- and tri-copper species, as well as larger copper oxide particles, have been proposed as active catalytic centers. The general trends in the stability of tetra- and pentamer copper oxide clusters (Cu_n_O_n_^2+^ and Cu_n_O_n−1_^2+^) stoichiometries embedded in the 8-ring channel of mordenite were studied in [58]. The authors showed that relative stability increases with cluster size. Aluminum content in mordenite crystal structure (Si/Al ratio) and its localization have a strong influence on cluster stability and its geometrical motif, which opens perspectives for tuning the properties of zeolites with copper oxo-clusters by changing this parameter for creating copper oxide clusters of a given structure and size. The effect of Si/Al ratio on the stability of silver clusters was previously discovered and studied in the range of Si/Al ratios from 5 to 103 [41,59]. There are similar studies on the properties of copper clusters [60,61]. A theoretical study of different copper oxide clusters in the main channel of mordenite was carried out on the basis of the calculations for periodic density functional theory [62].

The distribution of aluminum atoms in the crystal structure and its topology may have important implications for the formation of active and selective Cu-oxo compounds. Assessing the structure, formation pathways, and reactivity of mono- or multimeric Cu-oxo fragments presents a challenging platform for chemical scientists. Translating knowledge of Cu ion mobility and redox properties into the area of the direct conversion of methane to methanol may be important to better understand the redox properties of the transition metal ions in zeolites and to improve catalyst design and catalytic processes [63].

In this context, the search for alternative methods to synthesize Cu-oxo particles of different nucleations and structures placed within the MOR structure is an urgent task that paves the way for future applications and method development for Cu-zeolite research and beyond; to this context, this work contributes. Given the topological diversity of zeolites and the range of factors that influence the formation and stabilization of Cu-oxo-motifs, there are great opportunities to create such materials.

## 4. Conclusions

An Ag^+^-Cu^2+^-Zn^2+^/mordenite ternary system was prepared through microwave radiation-assisted ion-exchange treatment. Two sets of samples were prepared by a thermal reduction in hydrogen, followed by cooling in the air or in hydrogen were studied using XRD and UV-Vis spectroscopy. The XRD patterns show that the use of microwave radiation does not affect the structure of the parent mordenite. As a result of the ionic exchange and the position of the ionic species of these metals in certain cationic sites in the mordenite channels, the intensity of certain diffraction peaks in this zeolite was affected. This stands out for the (110) and (200) peaks, mainly for the latter, which is related to the extra-framework species sited in the mirror plane perpendicular to the a-axis, for which the intensity decreases with increasing silver content in the samples. This was associated with the fact that Ag^+^ ions occupy different cationic positions with respect to the starting Na^+^ cations in the mordenite. It was shown that, at 350 °C, the reduction of the trimetallic Ag^+^-Cu^2+^-Zn^2+^/mordenite systems with different metallic contents, followed by cooling in different atmospheres (air and hydrogen), leads to the obtainment of diverse metallic nanospecies. Copper and silver nanoparticles and Ag_8_ clusters were obtained for the hc samples reduced and cooled in hydrogen. A low silver content corresponds to the formation of a silver cluster, while the formation of nanoparticles of a bigger size took place in the trimetallic samples with higher silver content. No transformation of the Zn^2+^ cations was found. Cooling the reduced trimetallic samples in the air causes the oxidation of the clusters and silver and copper nanoparticles. This oxidation process is kinetically favored due to the high starting temperature of the cooling (350 °C). In the case of the copper nanoparticles, such conditions lead to the formation of mainly copper (II) oxide, while the silver nanospecies convert mainly into clusters and nanoparticles. The zinc ions present in the samples provoke changes in the mordenite matrix. The treatments used in this work (exchange, reduction in hydrogen, cooling in hydrogen, or cooling in the air) cause certain changes in the Zn-containing samples. The results suggest that the formation of point defects in the mordenite matrix and the formation of surface zinc oxide sub-nanoparticles in those samples reduced in hydrogen and then cooled in the air occurred. These results differ from those reported for the oxidation of samples reduced and cooled in hydrogen during oxidation under ambient conditions. In this latter case, the reduced nanospecies are re-oxidized to ionic species (e.g., Cu^2+^ cations) or nanospecies of a smaller size before the final, complete oxidation, and no evidence exists on the formation of metal oxides.

## Figures and Tables

**Figure 1 materials-16-00221-f001:**
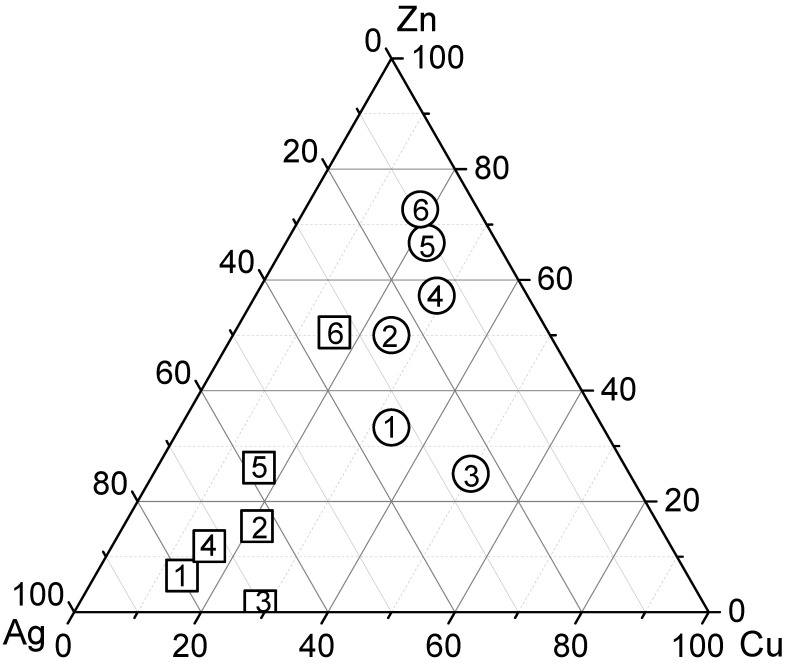
Ternary composition plot for the starting solutions (circles) and obtained samples (squares). Equal numbers denote the compositions for the initial solutions, as well as the samples of zeolites obtained using ion exchange treatment in these solutions: 1—stands for Ag1Cu1Zn1Mor, 2—for Ag1Cu1Zn2Mor, 3—for Ag1Cu2Zn1Mor, 4—for Ag1Cu2Zn4Mor, 5—for Ag1Cu2Zn6Mor, and 6—for Ag1Cu2Zn8Mor.

**Figure 2 materials-16-00221-f002:**
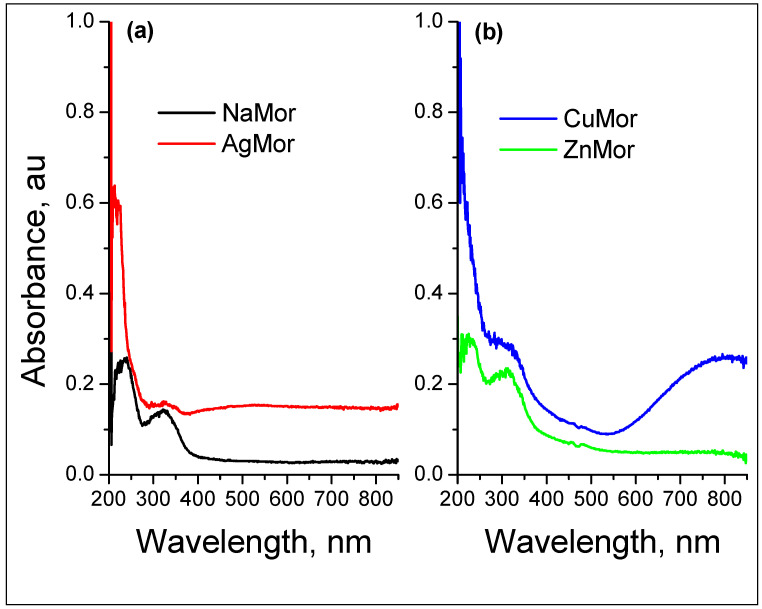
UV-Vis diffuse reflectance spectra of the NaMor and exchanged monometallic (**a**,**b**) and trimetallic (**c**,**d**) samples.

**Figure 3 materials-16-00221-f003:**
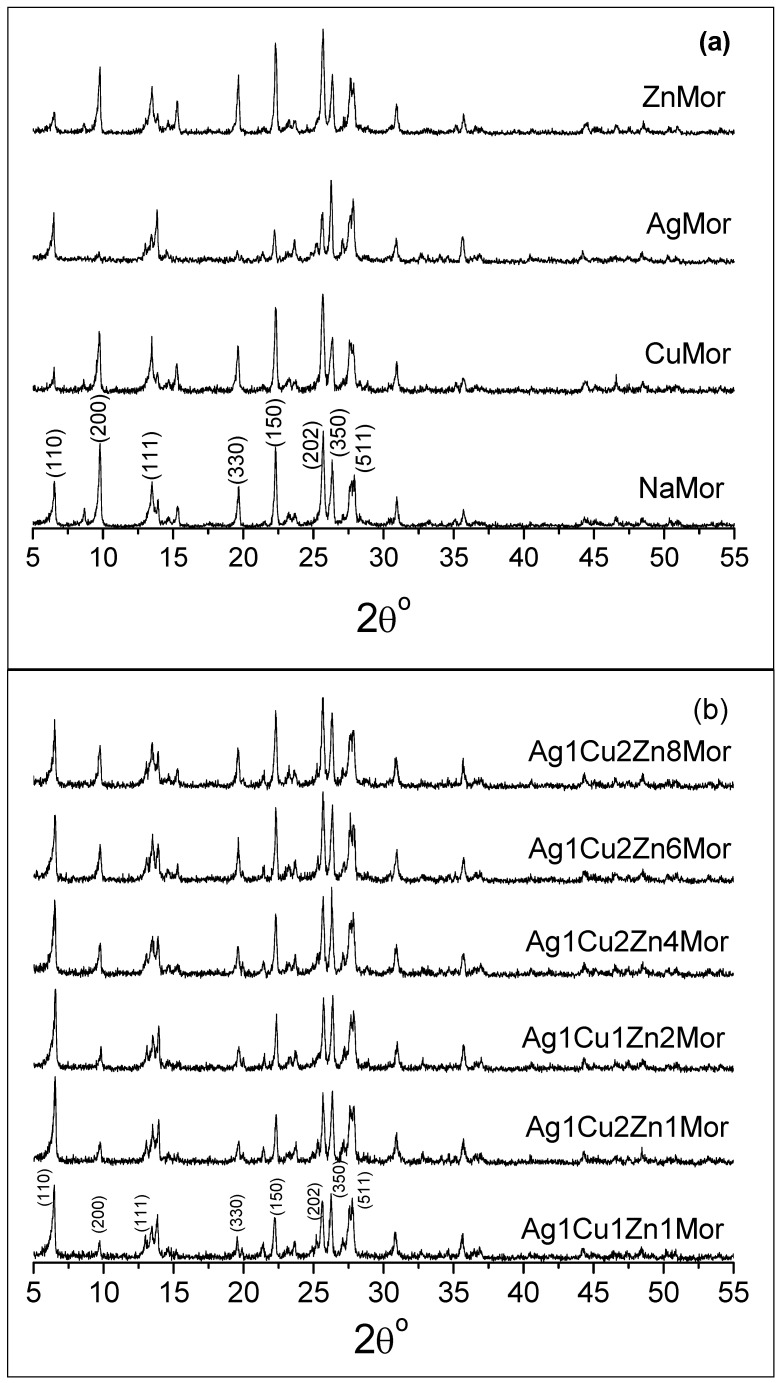
XRD patterns of the NaMor and exchanged monometallic and trimetallic samples. The Miller indexes of the main diffraction peaks for mordenite are shown.

**Figure 4 materials-16-00221-f004:**
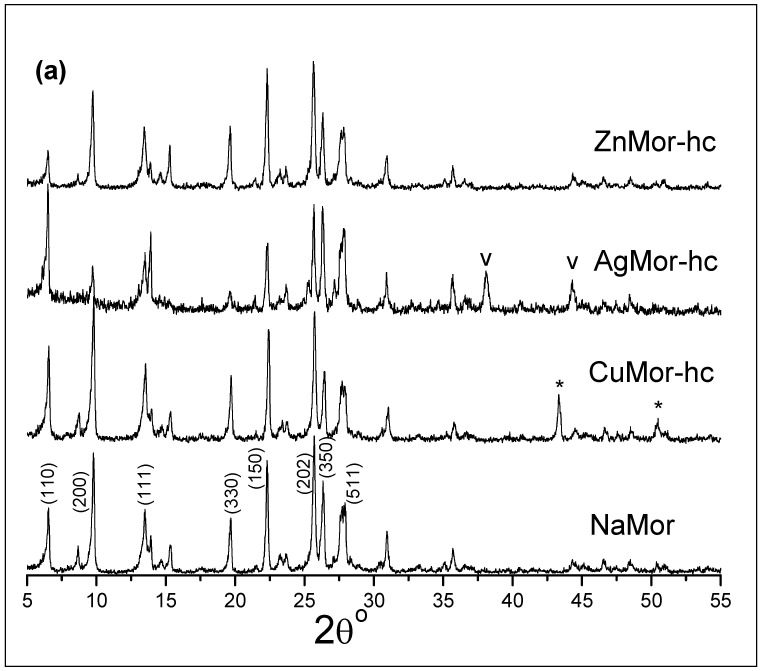
XRD patterns of the selected samples reduced in hydrogen and then also cooled in hydrogen. The peaks for metallic copper and silver are marked with asterisk (*) and “v” symbols, respectively. The pattern of the NaMor sample is included also.

**Figure 5 materials-16-00221-f005:**
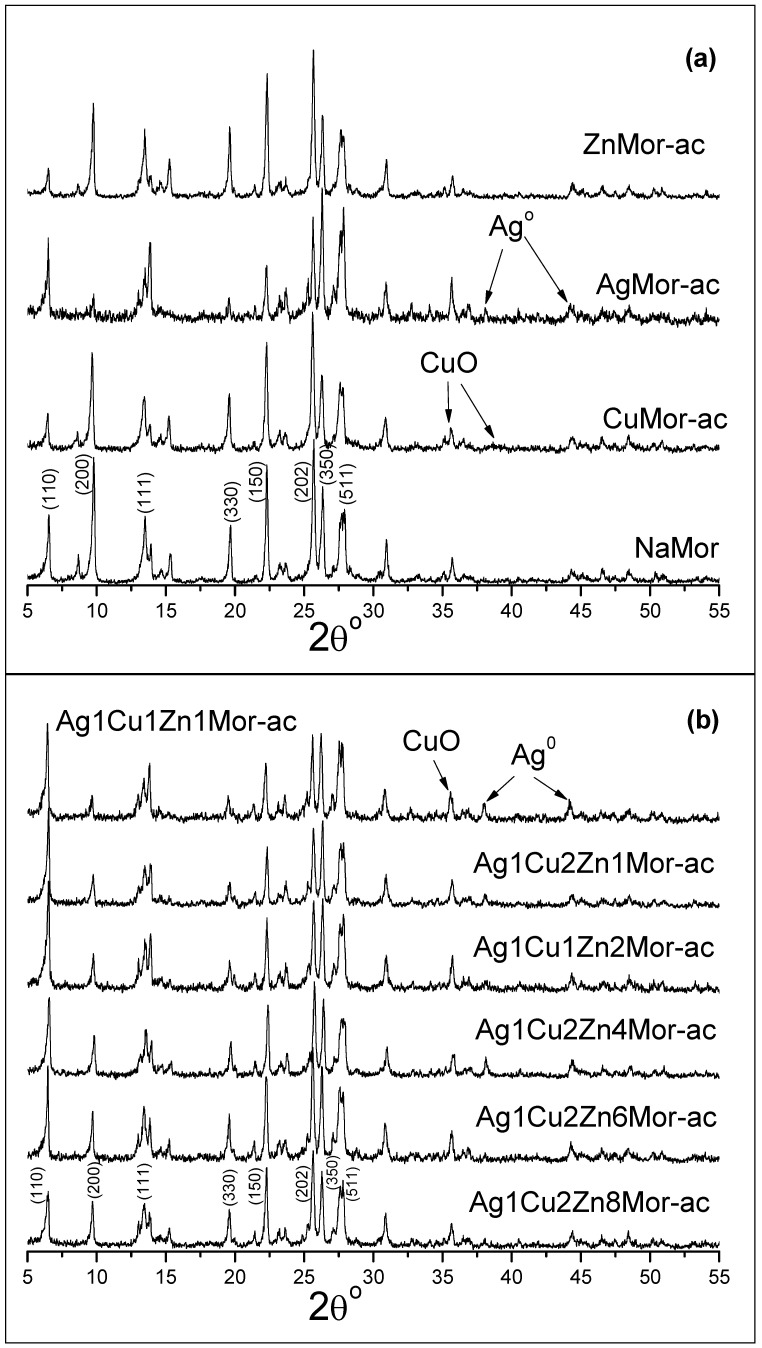
XRD patterns of the samples reduced in hydrogen, followed by cooling in air. The pattern of the NaMor sample is included also.

**Figure 6 materials-16-00221-f006:**
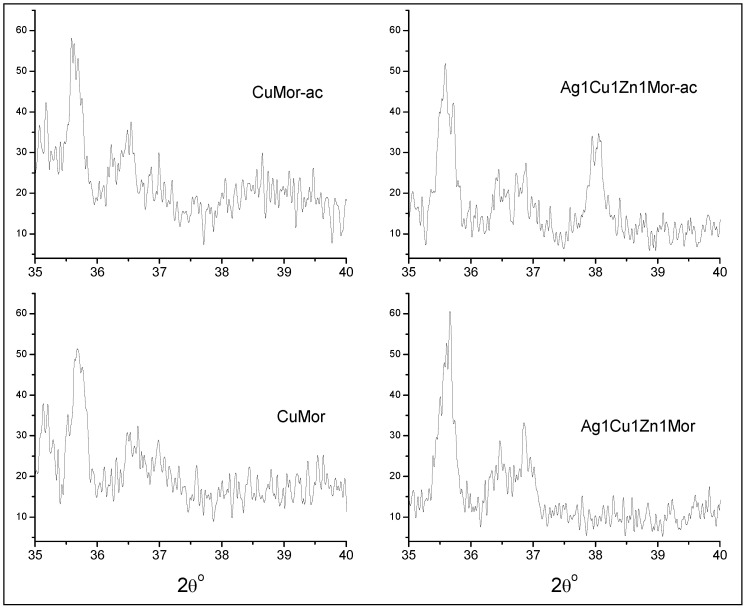
XRD patterns of 35–40° region of the selected samples: CuMor, CuMor-ac, Ag1Cu1Zn1Mor, and Ag1Cu1Zn1Mor-ac.

**Figure 7 materials-16-00221-f007:**
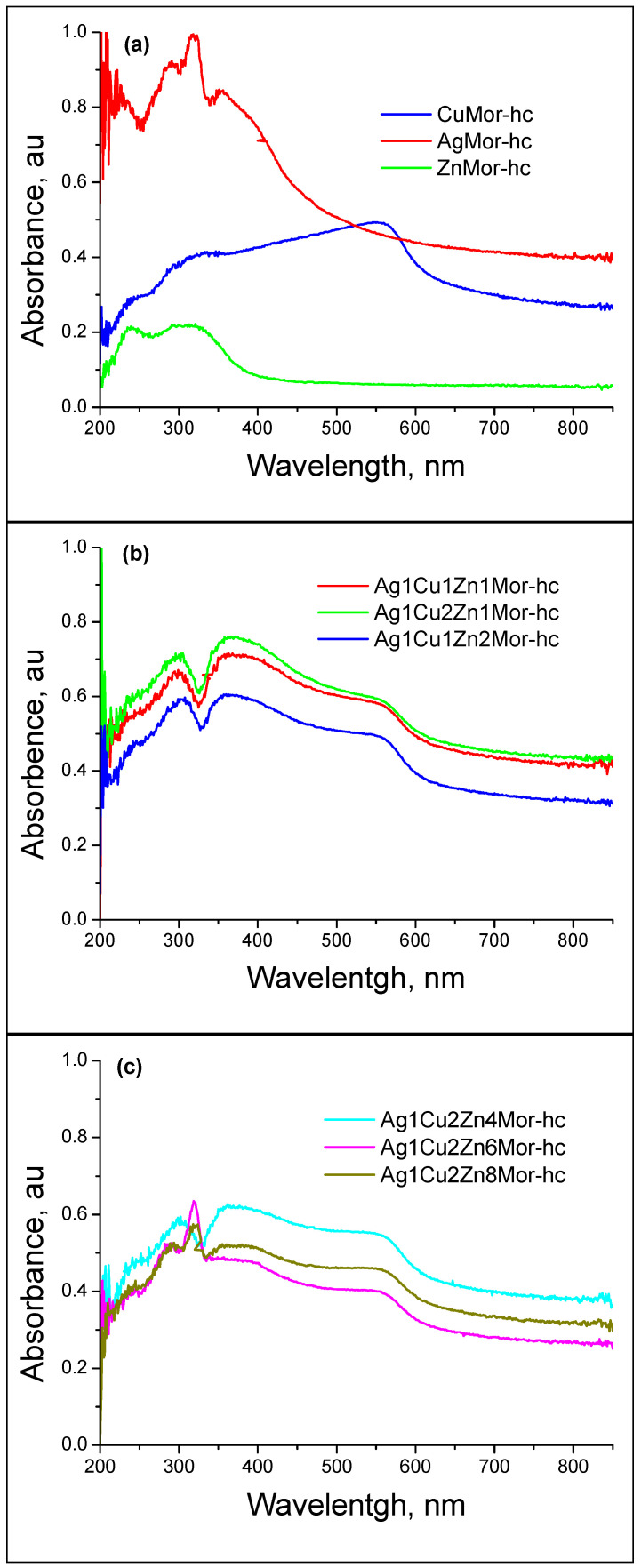
UV-Vis diffuse reflectance spectra of the samples reduced and then cooled in hydrogen.

**Figure 8 materials-16-00221-f008:**
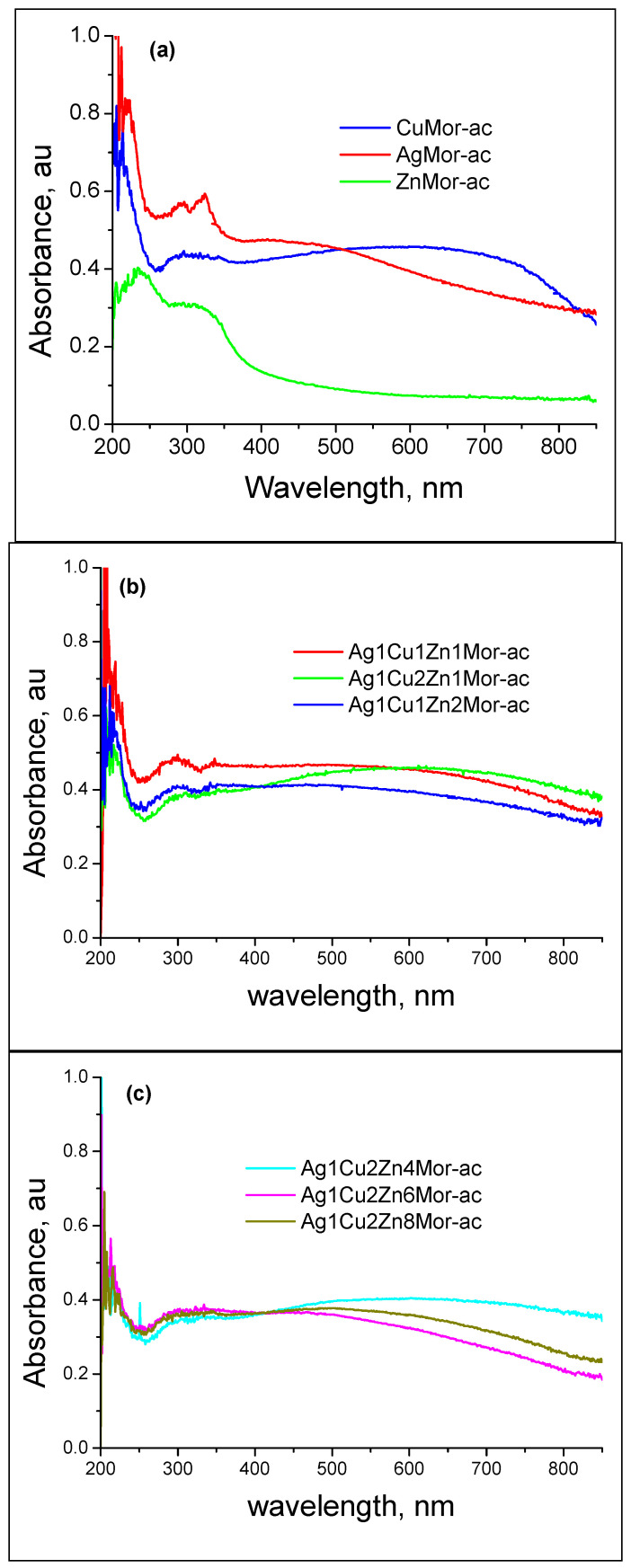
UV-Vis diffuse reflectance spectra of the samples reduced in hydrogen and then cooled in air.

**Figure 9 materials-16-00221-f009:**
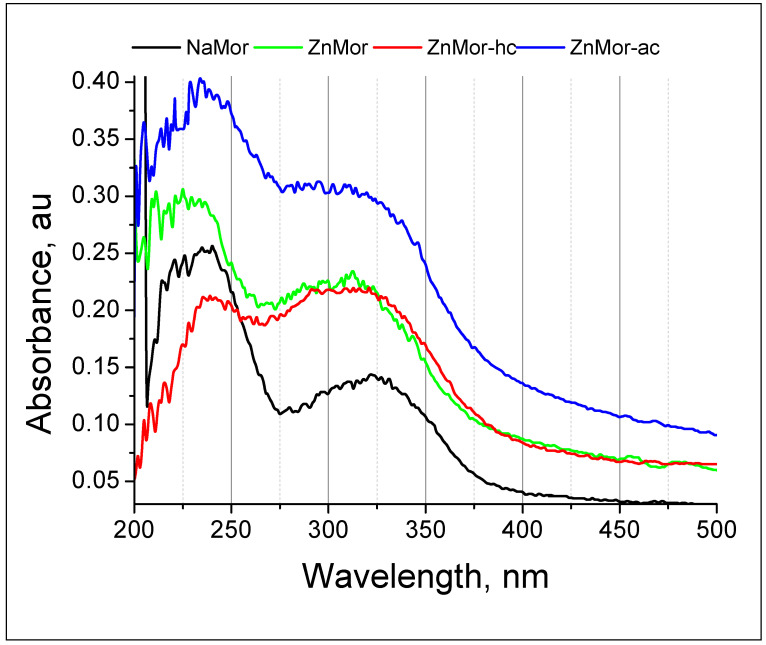
UV-Vis diffuse reflectance spectra of the 200–500 nm region of the selected samples: NaMor, ZnMor, ZnMor-hc and ZnMor-ac.

**Table 1 materials-16-00221-t001:** Silver, copper, and zinc contents (wt.%) for exchanged samples.

Samples	Ag	Cu	Zn
AgMor	12.75		
CuMor		4.07	
ZnMor			3.58
Ag1Cu1Zn1Mor	8.08	1.39	0.67
Ag1Cu1Zn2Mor	7.15	1.51	1.17
Ag1Cu2Zn1Mor	6.79	2.79	0.10
Ag1Cu2Zn4Mor	4.70	1.56	1.15
Ag1Cu2Zn6Mor	3.88	1.08	1.75
Ag1Cu2Zn8Mor	2.37	1.11	3.55

## Data Availability

Not applicable.

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
