# Peer review of "Mordenite-Supported Ag+-Cu2+-Zn2+ Trimetallic System: A Variety of Nanospecies Obtained via Thermal Reduction in Hydrogen Followed by Cooling in an Air or Hydrogen Atmosphere"

_materials, 2022, doi:10.3390/ma16010221_

Round 1

Reviewer 1 Report

1. There are a lot of language problems in the article, include tense, I hope the author can modify the language and typing problems in detail.

2. In the abstract, the authors should emphasize what results the characterizations indicate more specifically with crystalline parameters.

3. The temperature symbol is wrong in the manuscript authors have to verify it °C it must be like this.

4. authors must mention the sources for all the chemical and equipment’s used in the experiment.

5. authors must maintain one notation for y and x axis unit values like wavelength [nm] or just nm.

6. Figure 2 seems to be more wide spread which must be fitted perfectly by close packed box graph. With X’ and Y’ axis. Same as to Figure 6, 7, & 8.

7. In Fig 3A authors have mentioned h, k, l values only for NaMor it is requested to add values for all the samples which helps reader easily. Or indicate with a symbol showing the resemblance of the JCPDS or ICDS card numbers

8. In line 408 authors have mentioned about the surface of mordenite crystals it is requested to give a brief explanation how agglomeration has occurred with a specific reason.

9. In line 448 instead of statement authors are requested to make it as a point of explanation observed from the experiment which implies for many of the sentences from XRD and UV observations.

10. Authors must add recent studies and compare the outcome of the perspective with much deeper elaborations. Also, authors can site some recent publications,

Vilian, A. E., Veeramani, V., Chen, S. M., Madhu, R., Huh, Y. S., & Han, Y. K. (2015). Preparation of a reduced graphene oxide/poly-l-glutathione nanocomposite for electrochemical detection of 4-aminophenol in orange juice samples. Analytical Methods, 7(13), 5627-5634.

Hwa, K. Y., Ganguly, A., Santhan, A., & Sharma, T. S. K. (2022). Synthesis of Water-Soluble Cadmium Selenide/Zinc Sulfide Quantum Dots on Functionalized Multiwalled Carbon Nanotubes for Efficient Covalent Synergism in Determining Environmental Hazardous Phenolic Compounds. ACS Sustainable Chemistry & Engineering, 10(3), 1298-1315.

Jana, J., Van Phuc, T., Chung, J. S., Choi, W. M., & Hur, S. H. (2022). Nano-Dimensional Carbon Nanosphere Supported Non-Precious Metal Oxide Composite: A Cathode Material for Sea Water Reduction. Nanomaterials, 12(23), 4348.

Author Response

Please, find an attached file

Reviewer 2 Report

The manuscript is prepared in a professional way, but more can be done to improve its quality.

The authors supplied many data in this manuscript and some figures have many sets of data. I strongly advise the authors to include colour in these figures so as readers can easily differentiate the data.

Lack of key characterization. The description of Section 2.0 is TOO short (half page). Physical morphology of the key materials should be characterized using FESEM/TEM. Besides, the size of the key materials should be provided and be compared with other studies.

Title is too long. It should be brief but specific.

Abstract – Problem statement is not provided. Besides, key quantitative data is not given. Authors should also mention the applications for the developed materials.

Introduction – Authors cited too many references in this manuscript. For a simple sentence, there are 16 references cited (Line 41).

The problem statement and knowledge gap to be addressed are not clear. Please improve it.

Line 129 – Please provide the reasons on how these ratios were selected in the first place.

Figure 3 – Authors are advised to use “*” to indicate the key differences of the data.

Comparison table – Please come out with a table to compare side-by-side the properties of the developed materials with other relevant studies.

Conclusion - Key quantitative data should be provided.

Author Response

Please, find an attached file

Reviewer 3 Report

The manuscript showed the study of thermal reduction of ternary Ag+-Cu2+-Zn2+/MOR system in hydrogen and followed by cooling in air or hydrogen. Overall, the manuscript is well-written, which should be published after minor revision. The comments are as follows.

1.     The title is too long, and the authors should give a concise one.

2.     Why use the MOR zeolite? What about others zeolite such as FAU, Beta or ZSM-5 zeolite?

3.     The authors choose the AgNO3, Cu(NO3)2, and Zn(NO3)2 in this work. What about other metal salts with different anion?

Author Response

Please, find an attached file

Reviewer 4 Report

In this work, the authors have synthesized trimetallic Ag+-Cu2+-Zn2+ modified Mordenite zeolites and characterized the metallic nano-species formed during thermal reduction in hydrogen with two different cooling processes. Their results demonstrate the synergistic interactions among the multimetallic systems. This manuscript is very well written. This work is of interest to the audience of Materials.  There are just few minor concerns, though, that need to be addressed before acceptance.

1.     The authors mentioned that thermal reduction in bimetallic Cu/Ag and Cu/Zn systems have been studied previously. Is it possible that the authors can compare the UV-Vis spectra of those bimetallic systems, if available under the similar experimental conditions, to Figure 8 and 9 to provide some insight on how the 3rd co-ion can affect the original bimetallic systems?

2.     Table 1: I wonder why the Zn content is so low (only 0.1 wt. %) in the Ag1Cu2Zn1Mor sample. Comparing the two samples Ag1Cu1Zn1Mor and Ag1Cu2Zn1Mor, the Cu content is increased by a factor of ~2, whereas the Zn content is decreased by a factor of ~6. What is the error bar in this measurement?

Author Response

Please, find an attached file

Round 2

Reviewer 2 Report

The authors have responded well to the concerns raised.